# Container Shipping Optimization under Different Carbon Emission Policies: A Case Study

**Xiangang Lan** [1], **Xiaode Zuo** [1,*] and **Qin Tao** [2]

1. School of Management, Jinan University, Guangzhou 510632, China
2. Faculty of Business, City University of Macau, Macau, China
* Correspondence: tzuoxd@jnu.edu.cn

**Abstract:** Climate change is a major environmental issue facing humanity today, and the International Maritime Organization has accelerated the formulation of greenhouse gas emission policies. This study considers different carbon emission policies to construct an optimization model for container shipping, design an improved Whale Swarm Algorithm to solve related issues, and use the marginal carbon abatement cost method to analyze the deep-seated reasons for the optimization of liner shipping according to different carbon emission policies, thereby revealing the underlying reasons of emission-reduction decisions. The conclusions reveal that both kinds of carbon emission policies will reduce the profits of companies, the average speed of shipping, and carbon emissions. The carbon tax model has the greatest impact on the profits of shipping companies, and carbon cap-and-trade is easier to obtain support from enterprises. Sensitivity analysis shows that the implementation of carbon cap-and-trade or a carbon tax policy is closely and complexly related to the carbon trading price, carbon tax rate, fuel price, and ship size, and there is uncertainty.

**Keywords:** carbon emission policies; container shipping optimization; marginal carbon abatement cost method; Whale Swarm Algorithm

## 1. Introduction

Maritime transport is responsible for more than 80% of the freight volume of international trade [1], with container liner shipping accounting for 24% of global maritime business. In recent decades, the pollution problem of the shipping industry has attracted increasing attention. The fourth Greenhouse Gas Study was released in August 2020 (by the International Maritime Organization) [2], which reported that global $CO_2$ emissions from shipping amounted to 794 million tons in 2020, accounting for 3.1% of the world's total $CO_2$ emissions. Climate change is a major environmental issue facing humanity today. By 2030, global average temperatures are expected to be 1.5 degrees warmer than before the Industrial Revolution or even higher [3]. Therefore, it is particularly important to transition the shipping industry toward an environmentally sustainable mode of development. The IMO has accelerated the development of a greenhouse gas emissions policy, and in 2018, it proposed its first greenhouse gas reduction strategy. By the middle of the century, global shipping carbon dioxide emissions will be cut by 50 percent from 2008 levels and gradually move closer to zero carbon emissions [4]. With the end of the COVID-19 pandemic, the shipping market has returned to a downturn; simultaneously, the IMO is accelerating the formulation of greenhouse gas emissions policies, with shipping gradually transitioning toward the goal of zero carbon emissions. This renders it even more necessary to optimize transportation based on analyzing the internal mechanism of emissions policies to provide countermeasures and suggestions for companies and policymakers. There are two main potential carbon emissions policies available: the carbon tax and carbon cap-and-trade [5–7]. In this context, this paper compares the regulation effects of two common carbon emission policies, as well as enterprise transportation optimization strategies under different regulation policies. Xing [5] studied the fleet configuration and speed optimization of container

liners based on two carbon emission policies (carbon tax and carbon cap-and-trade). Some interesting conclusions have also been drawn from related studies, which provide the research basis for this study.

However, few scholars have revealed the internal mechanism of the two emission policies by comparing them and analyzing their marginal costs, adaptability, and advantages and disadvantages. In response to these two carbon emission adjustment policies, what kind of transportation strategy should companies adopt, and how should they optimize transportation in a timely manner considering market changes and policy adjustments? Against the backdrop of green shipping, it is necessary to consider protecting the environment without affecting the healthy development of the shipping market. How, then, should policymakers choose the optimal policy? To solve these problems, we propose the following research objectives:

- Profit maximum, type selection, ship number, and speed as decision variables, build the two common carbon emissions under the policy of the liner optimization model and design the algorithm to solve.
- Through empirical analysis, sensitivity analysis, and marginal cost analysis of common carbon emission adjustment policies, the internal mechanism of emission policies is revealed, and the regulation effect, adaptability, and advantages and disadvantages of the two kinds of carbon emission policies are analyzed.
- According to the analysis results, it provides reference suggestions for transportation optimization of enterprises and policy formulation of relevant departments.

To achieve the above goals, we incorporated the cost of carbon emissions into the cost structure of marine transportation. On an intercontinental container liner route comprising multiple ports, the shipping company provides liner transportation services for cargo owners according to a pre-defined schedule that includes ports of call and their sequences and charges for freight according to the liner tariff. When the shipping company's supply of goods is stable, and the freight rate is stable, it can be considered that its transportation revenue is unchanged, and profit is equal to revenue minus cost. The nonlinear optimization model of shipping speed with a maximum profit was constructed by taking the choice of vessel type, input quantity of vessels, and speed as decision variables. An improved Whale Swarm Algorithm for solving the model was designed, and real data were collected for a real case analysis. The results of the empirical analysis were used to compare the two policies' controlling of carbon emissions and analyze the impact of fluctuations in the carbon tax, carbon price, and fuel price on transportation optimization and profit. Finally, the marginal carbon abatement cost method was used to analyze the impact of both types of carbon emission policies on the optimization of liner shipping and reveal the internal mechanism of emission reduction decisions.

The remainder of this paper is organized as follows. Section 2 presents a literature review. The models are presented in Section 3. The proposed algorithm is designed in Section 4. The case study is presented in Section 5. Marginal abatement cost analysis and summary conclusions are presented in Sections 6 and 7, respectively.

## 2. Literature Review

### 2.1. Liner Transportation Optimization

Kontovas C. [8] introduced the concept, model, and scenarios suitable for speed optimization. Fagerholt [9] set up a ship speed optimization model for a route composed of a series of ports with the goal of minimizing fuel consumption, optimized the speed of each route to achieve a large amount of cost savings, and used heuristic algorithms to solve the problem. Shuaian Wang [10] calibrated the functional relationship between fuel consumption and speed. Using the historical data of a global shipping company for regression analysis, the cubed functional relationship between fuel consumption and speed was found to be relatively accurate. Ellen [11] studied optimal speed on the basis of reducing fuel consumption and emissions without increasing the number of ships in the fleet. Wang [12] studied the determination of refueling ports and the optimization of sailing speed under

different oil prices. Wang [13] set revenue maximization as the goal, considered the influence of containers of different sizes and seasonal changes in transportation demand, and built a liner transportation revenue management model to reasonably optimize shipping speed, reduce fuel consumption, and increase revenue. Guericke [14] constructed a cargo distribution model for container liner transport considering speed optimization and service level. Norlund [15] studied ship speed optimization under different weather conditions and built simulation models to estimate the weekly average fuel consumption of ships. Wen [16] studied ship navigation path and speed selection from multiple perspectives, such as voyage time, cost, and environment. Maricruz [17] built a discrete simulation model to study ship deceleration of dry bulk cargo fleet considering normal speed, deceleration route, and ultra-low speed sailing. Adland [18] studied the selection of dynamic speed for bulk shipping by taking into account factors such as organizational constraints, crew quality, trading mode, loading conditions, technical constraints, and special ship type with 18,000 voyage data collected from AIS. Karsten [19] built a comprehensive optimization model considering the sailing speed, ship path, and voyage time limits of container ships. Corbett [20] proposed the problem of $CO_2$ emission and ship speed optimization. Taking voyage profit maximization as the objective function, they studied the influence of ship deceleration on $CO_2$ emission under fixed routes. Zheng [21] addresses single-line shipping services for ship speed and fleet size optimization. Zhao [22] proposed an optimization model considering the minimum operation cost of the fleet (including voyage cost, operation cost, and capital cost). Speed is a key determinant of fuel cost. Adding just a few knots can lead to a dramatic increase in fuel consumption. Speed has a significant impact on operating costs. Speed optimization has been a hot issue in the liner shipping research field for the last ten years, and scholars have performed a lot of research in this aspect. These studies provide a solid foundation for our study, but most scholars only take the speed as the decision variable and use the algorithm to solve it; few studies consider the selection of ship type, ship investment, and speed optimization as the decision variables from the perspective of liner shipping companies operating a route.

### 2.2. Liner Transportation Optimization Considering Carbon Emission Regulation Policies

Kim [23] builds a model to determine the fleet size, ship speed, and the number of charters according to maritime environmental regulations (including carbon tax and carbon emission trading). Lee [24] studied the influence of carbon tax policy (GTAP-E) on container shipping. Huang [25] proposed a ship emission trading scheme with $CO_2$ as its main content and the impact of $CO_2$ emission trading based on the deceleration of annual profits and $CO_2$ emissions of container ships. Cullinane [26] summarized the current policy system of shipping carbon emissions and believed that the combination of policy and technological innovation could better promote the emission reduction of ships. Kim [27] studied the speed optimization of ships on routes with designated call port sequences by considering environmental regulations to reduce carbon emissions and fuel consumption. Wang [28] established three types of carbon emission tax (no carbon emission tax, limited carbon emission tax, and carbon emission tax) voyage speed decision model. Wang [29] studied the impact of carbon tax policy on ship refueling strategy, speed, and deployment. Zhu [30] studied fleet planning under the uncertainty of carbon tax policy by a multi-stage stochastic integer programming model. Cheaitou [31] proposed a multi-objective optimization model of liner transportation based on $CO_2$ emission minimization, $SO_x$ emission minimization, and profit maximization in which all objective functions of the model are functions of ship sailing speed. Xin et al. [32], considering the characteristics of modern crude oil supply systems, discussed the green scheduling problem of tanker fleets considering variable speed and carbon tax.

The shipping industry was relatively prosperous from 2000 to 2007, especially in 2007, which represented a golden period of shipping development, with fuel prices for ships being significantly lower than today's prices. Therefore, to maximize profits, most ships have adopted high-speed sailing; hence, relatively limited literature has addressed

speed optimization. With the impact of the 2008 financial crisis, the shipping market has undergone a 180-degree shift, and coupled with the rising fuel prices, shipping companies have adopted low-speed sailing, resulting in an increasing amount of literature related to shipping transportation optimization. Xing et al. [33] sorted out a series of methods to reduce shipping carbon emissions. In addition to the widely studied technical and operational measures, they also discussed market-level measures and independent emission reduction behaviors. MA et al. [34] carried out multi-objective optimization of route and speed under weather changes and emission control policies. LAN [35] consider carbon emission and emission control area built a liner transport network design model, including route design, route allocation, and cargo transport plan.

Carbon emission reduction has become an inevitable trend in the development of the shipping industry. In order to reduce the carbon emissions of the shipping industry, the international community has put forward different carbon emission policies. Carbon tax and carbon emission quota are the two main carbon emission policies. Therefore, carbon tax policy is bound to have an impact on transportation optimization decisions. Most of the existing relevant studies only focus on the impact of carbon emission policies on ship speed and only consider carbon emission regulation policies to study the optimal speed of traditional transportation optimization, while few scholars reveal the internal mechanism of emissions policies and analyze their adaptability and advantages by comparing common carbon emission adjustment policies and conducting marginal cost analysis. This study, based on the original research, takes ship type selection, ship input quantity, and speed as decision variables and considers different carbon emission regulation policies to build an optimization model for container liner transportation. Through empirical analysis, sensitivity analysis, and marginal cost analysis of common carbon emission regulation policies, the internal mechanism of emission policies is revealed. The regulation effect, adaptability, and pros and cons of the two carbon emission policies are analyzed.

## 3. Models

On an intercontinental container liner route, the shipping company provides transportation services for cargo owners according to a pre-defined schedule, including ports of call and their sequence and charges for freight according to the liner tariff. Profit is equal to revenue minus costs, which include fixed vessel operating costs, main engine and auxiliary engine fuel costs, carbon emissions costs, and port costs. A nonlinear optimization model of shipping speed with the maximum profit of the shipping company is constructed by taking the speed, input quantity of vessels, and choice of vessel type as decision variables.

Assumptions: (1) Excess transport capacity, regardless of ship input cost; (2) No special circumstances shall be taken into account in the normal navigation of the ship, and no accident shall occur in the course of navigation; (3) The ship will operate in the form of return voyage in the past, with weekly shift frequency for each route and the same speed for the same route; (4) During the study period, the demand for cargo transportation was stable, and the freight volume between ports remained basically unchanged; (5) The size of the ship type on the route is determined by the freight volume, while the freight volume does not change, so the ship type invested in the same route is the same, but the ship type selection of the route is decided by model optimization; (6) The infrastructure of all ports has been mature, assuming that the loading and unloading efficiency, charging standard, ship arrival and departure time of all ports selected in this study are the same.

$k$ denotes the shipping segment, and $Y_k$ denotes the total amount of cargo (tons) on segment $k$. $q_{ij}$ represents a container transport from port $i$ to port $j$, as presented in Figure 1: When port 2 has container $q_{12}$ to be transported to port 4, it needs to pass through Sections 2 and 3. When port 1 has container $q_{13}$ to be transported to port 3, it must pass through Sections 1 and 2. In that case, the total number of containers transported on segment 3 is $q_{24} + q_{13}$. Similarly, the cargo volume $Y_k$ on segment $k$ is denoted as $Y_k = \sum_{j=k+1}^{N} \sum_{i=1}^{k} q_{ij}$. Figure 1 represents the round-trip route. As presented in Figure 2,

the export route is (to): Ningbo Port—Shanghai Port—Busan Port—Long Beach Port. The import route is (back): Long Beach Port—Busan Port—Ningbo Port when $N = 6$.

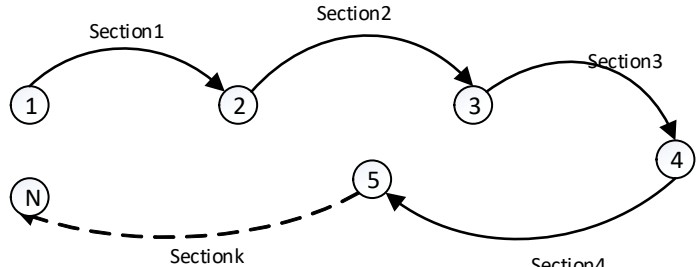

**Figure 1.** Relationship between the cargo volume and the OD demand between ports.

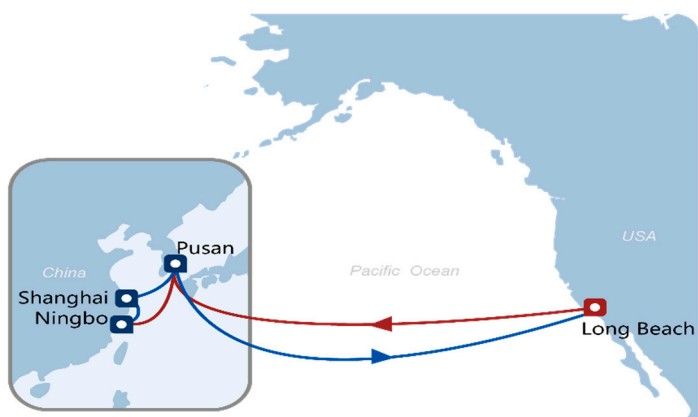

**Figure 2.** COSCO Shipping Trans-Pacific Service—AAC4.

The total time $T$ of a round trip includes sea travel time, arrival and departure time, and loading and unloading time. $Q_i$ is the loading and unloading of port $i$ $Q_i = \sum_{j=1}^{N}(q_{ij} + q_{ji})$ the expression of the total voyage time $T$ is:

$$T = D/v + \sum_{i=1}^{N} Q_i/l + N \cdot t_{pil} \tag{1}$$

Suppose that the number of ships per week of the route is $n_m$, and the total number of ships per week is 168 h, then the expression of the number of ships per week $n_m$ is:

$$n_m = \frac{T}{168} = \frac{1}{168}\left(D/\sum_{m \in M}(x_m v_m) + \sum_{i=1}^{N} Q_i/l + N \cdot t_{pil}\right) \tag{2}$$

Suppose the daily revenue of the fleet is $f(r)$, the daily revenue and freight rate multiplied by the weekly volume divided by 7, and the expression is:

$$f(r) = \sum_{i=0}^{N-1} \sum_{j=0}^{N-1} q_{ij} r_{ij}/7 \tag{3}$$

The average daily total cost of the fleet includes port charges, fuel consumption charges, and operating costs. Let the weekly operating costs of the fleet be $C^{op}$, and the average port charges per week be $C^P$ (Including port usage fee, mooring fee, berthing fee, and other fixed charges). Then, the expressions $C^{op}$, $C^P$:

$$C^{op} = \sum_{m \in M}(x_m C_m n_m) \tag{4}$$

$$C^P = \sum_{i=1}^{N} \sum_{j=1}^{N} (q_{ij}C^l + q_{ji}C^u)/7 + N \cdot \sum_{i=1}^{N} (G_m^0 x_m)/7 \tag{5}$$

According to the cubed function between speed and fuel consumption [10] $F(v/v_0)^3$. The main engine uses heavy oil, and the auxiliary engine uses light oil. Then, the expression $F$.

$$F = \sum_{m \in M} x_m F_m \left(\frac{v_m}{v_m^0}\right)^3 \frac{D}{168 v_m} + \sum_{m \in M} A_m \cdot x_m \cdot n_m \tag{6}$$

Ship carbon emissions depend on ship fuel consumption in a certain period and the carbon emission factor of fuel $\lambda$. According to Fourth Greenhouse Gas Study 2020, this paper uses the factor $\lambda = 3.114$. That is, $1t$ of marine fuel produces $3.114t$ of $CO_2$. $Q_{co_2}$ can be expressed as:

$$Q_{co_2} = \lambda \cdot F_0 \left(\frac{v}{v_0}\right)^3 \frac{D}{168v} + \lambda \cdot A \cdot n \tag{7}$$

### 3.1. Optimization Model of Liner Shipping under a Carbon Tax Policy

Carbon tax refers to the policy of imposing a certain tax on shipping enterprises according to their carbon emissions. Volume-based carbon taxes are levied based on the ship's $CO_2$ emissions, assuming that a quantitative carbon tax $\delta$ is collected per ton (USD\$/ton), and the carbon emission cost $C_{co_2}$ is expressed as (8):

$$C_{co_2} = Q_{co_2} \cdot \delta \tag{8}$$

Objective function:

$$\max f^{day}(tp) = \sum_{i=1}^{N} \sum_{j=1}^{N} \frac{q_{ij} \cdot r_{ij}}{7} - \sum_{m \in M} x_m F_m \left(\frac{v_m}{v_m^0}\right)^3 \cdot \frac{D}{v_m \cdot 168} \cdot P_{HFO} - \sum_{m \in M} A_m \cdot x_m \cdot n_m \cdot P_{MFO}$$
$$- \sum_{m \in M} (x_m C_m n_m) - \sum_{i=1}^{N} \sum_{j=1}^{N} (q_{ij}C^l + q_{ji}C^u)/7 - \sum_{m \in M} (x_m G_m^0) \cdot N/7 - Q_{co_2} \cdot \delta \tag{9}$$

Constraints:

$$\sum_{m \in M} x_m = 1, \ x_m = 0 \text{ or } 1 \tag{10}$$

$$\max(Y_k) \le \sum_{m \in M} B_m x_m \tag{11}$$

$$v_m^{min} \le v_m \le v_m^{max} \tag{12}$$

$$k = 1, 2, \cdots, N-1 \quad i, j = 1, 2, \cdots, N \tag{13}$$

$$m \in M \tag{14}$$

With the vessel speed, choice of vessel type, and input quantity of vessels as the decision variables, the objective function (9) has two parts. The first component is the freight revenue, while the second component is the cost, including the average daily fuel cost of the main engine, the average daily fuel cost of the auxiliary engine, the average daily operating cost (daily rent of the vessel), loading and unloading fees, port fees, and carbon emission cost. (10) The constraint is that ships of the same type are deployed on the route. (11) The constraint is the vessel capacity restriction. (12) The constraint is the vessel speed limits. (13) This includes non-negative and integer constraints. (14) The constraint is the vessel type.

*3.2. Optimization Model of Liner Transportation under the Carbon Cap-and-Trade Policy*

The carbon allowance is the production and operation of enterprises in accordance with the limited carbon emission quota. Companies are not allowed to enter the carbon cap-and-trade market to buy or sell carbon credits when carbon emission quotas are insufficient or excessive. Therefore, the carbon allowance policy is a mandatory constraint for manufacturing companies, assuming that carbon emissions are limited to no more than $\theta$ (tons). Based on the above analysis of each cost item, the optimization model of liner shipping under the multi-vessel carbon cap-and-trade policy is expressed as follows:

Objective function:

$$
\begin{aligned}
\max f^{day}(tp) \quad &= \sum_{i=1}^{N}\sum_{j=1}^{N}\frac{q_{ij}\cdot r_{ij}}{7} - \sum_{m\in M} x_m F_m\left(\frac{v_m}{v_m^0}\right)^3 \cdot \frac{D}{v_m\cdot 168}\cdot P_{HFO} - \sum_{m\in M} A_m\cdot x_m\cdot n_m\cdot P_{MFO} \\
&- \sum_{m\in M}(x_m C_m n_m) - \sum_{i=1}^{N}\sum_{j=1}^{N}\frac{q_{ij}C^l+q_{ji}C^u}{7} - \sum_{m\in M}\frac{x_m G_m^0 N}{7} - (Q_{co_2}-\theta)\cdot p_e
\end{aligned}
$$

(15)

Constraints: (10)–(14)

With speed, choice of vessel type, and input quantity of vessels as the decision variables, the objective function (16) has two parts. The first component is the freight revenue, while the second component is the cost, including the average daily fuel cost of the main engine, the average daily fuel cost of the auxiliary engine, the average daily operating cost (daily rent of the vessel), loading and unloading fees, port fees, and carbon emission cost.

**4. Arithmetic Design**

Using the MATLAB platform, an improved Whale Swam Algorithm (WSA) was designed to solve the planning model. Swarm behaviors, such as predation of the whale swarm with ultrasonic waves as the information medium, are simulated, and each solution is compared to a whale. The movement of each whale is guided by the nearest whale among the whales that are better than it (judged by their fitness value), and this leading whale is defined as the "better and nearest" whale.

The flow chart of the algorithm is presented in Figure 3.

*Description of Improved Whale Swarm Algorithm*

(1) The intensity of the ultrasonic source decreases linearly $rho0 = rh\max - \frac{iter\cdot(rh\max - rh\min)}{iterations}$.

(2) normalization: As the range of independent variables in this study is considerably large and performing optimization in a large range is significantly more difficult, this algorithm will normalize the independent variables. For example, an individual whale is $pop_{j,k}$, where $pop_{j,k} \in [0,1]$, and its corresponding actual value is $(\max value - \min value)\cdot pop_i + \min value$.

(3) The ultrasonic attenuation coefficient. The intensity of the ultrasonic source is attenuated as a multiple of the attenuation coefficient, and the attenuation system formula is $eta = -\log\left(\frac{0.25}{Farthest\_fit}\right)$ where $Farthest\_fit$ is the fitness value of the farthest individual from the $j$-th whale and the intensity of the ultrasonic source is $rho0 = eta\cdot rho0$.

(4) The improved population update method. If the nearest individual is better than the current individual, the individual is updated according to the formula $pop_j = pop_j + rho\cdot rand()\cdot(Nearest\_pop - pop_j)$. If it is worse, the new individual is generated in reverse $newpop_j = pop_j - rho\cdot rand()\cdot(Nearest\_pop - pop_j)$, and if the new individual is better than the current individual, it replaces the latter; otherwise, the original individual is retained.

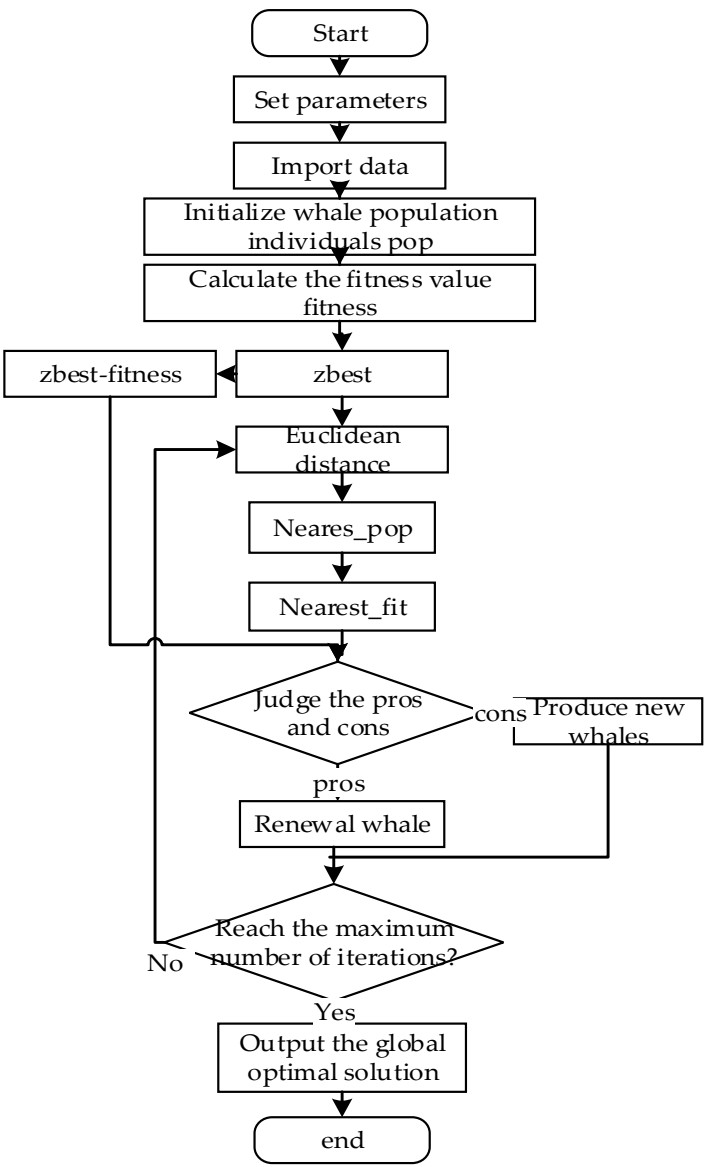

**Figure 3.** Flow chart of improved Whale Swarm Algorithm.

## 5. Case Study

*5.1. Data Collection*

In order to improve the research significance, this paper studies the real case of the COSCO Shipping Container Lines Co., Ltd., Shanghai, China, as shown in Figure 4. This study assumes that the ship departs from and returns to the Port of Shanghai. Its western departure is as follows: Port of Shanghai—Port of Ningbo—Port of Xiamen—Port of Yantian—Port of Singapore—Port of Felixstowe; it returns toward the east as follows: Port of Rotterdam—Port of Wilhelmshaven—Port of Gdansk—Port of Felixstowe—Port of Singapore—Port of Yantian—Port of Shanghai; D = 2483 kn, N = 13, and Hong Kong Feb 15 prices in USD are as follows: $P_{IFO}$ = 426.5 USD/TON, $P_{VLSFO}$ = 628 USD/TON; According to data from Yantian Port: l = 150 TEU/h, $C^l = C^u$ = 65 USD/TEU, $t_{pil}$ = 2 h; reference ESMA Final Report [36]: $\partial$ = 63 !USD/TON, $\theta$ = 1600 Ton/day; accord to Lowe's Marine Database, data of each ship type are shown in Table 1.

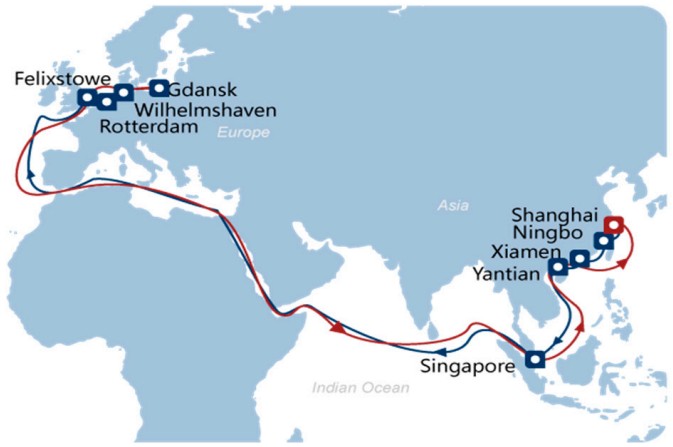

**Figure 4.** COSCO Far East—Mediterranean route chart.

**Table 1.** Data of each ship type.

| Name of Vessel | m | Bm (TEU) | $F_m$ (T/day) | $A_m$ (T/Day) | $v_m^0$ (Kn/h) | $v_m^{min}$ (Kn/h) | $v_m^{max}$ (Kn/h) | $c_m$ (USD/Day) | $G_m^0$ (USD/Call) |
|---|---|---|---|---|---|---|---|---|---|
| XIN ZHANG ZHOU | 1 | 4253 | 139.5 | 6.33 | 18.2 | 11.34 | 25.15 | 9000 | 3001 |
| XIN WENZHOU | 2 | 4738 | 82 | 4.3 | 18 | 11.04 | 24.7 | 10,026 | 3344 |
| XIN YAN TIAN | 3 | 5668 | 202 | 7.81 | 17.7 | 12.05 | 26.7 | 11,994 | 4000 |
| COSCO THAILAND | 4 | 8501 | 250 | 10.47 | 18.6 | 12 | 26.6 | 17,989 | 6000 |
| XIN SHANGHAI | 5 | 9572 | 248.2 | 10.43 | 17.2 | 11.22 | 26.73 | 20,255 | 6204 |
| COSCO ASIA | 6 | 10,036 | 250 | 12.75 | 16.8 | 11.04 | 25.8 | 21,238 | 6505 |
| COSCO FAITH | 7 | 13,114 | 274.9 | 13.2 | 16.7 | 11 | 26.2 | 27,751 | 8500 |
| CSCLJUPITER | 8 | 14,074 | 262 | 14.51 | 16.1 | 11.18 | 26.62 | 29,783 | 9122 |
| CSCLPACIFIC OCEAN | 9 | 18,982 | 195.5 | 13.768 | 18 | 10 | 24.6 | 40,169 | 13,000 |
| COSCO SHIPPING VIRGO | 10 | 20,119 | 168 | 10.263 | 19 | 8.4615 | 22.5 | 42,575 | 13,040 |

Data source: Lowe's Marine Database.

Drewry Shipping Consultant The weekly demand from Asian ports to a European port (westbound) is about 1000 TEU. Drewry Shipping Consultant The weekly demand from Asian ports to a European port (westbound) is about 1000 TEU.inter-port rate according to China International Shipping Network, See Table 2.

**Table 2.** Interport traffic and freight rate (TEU, freight rate USD/TEU).

| Outward Voyage | Singapore | Felixstowe | Rotterdam |
|---|---|---|---|
| Shanghai | (570,150) | (1040,800) | (1040,850) |
| Ningbo | (570,150) | (1050,750) | (1050,750) |
| Xiamen | (380,140) | (1040,745) | (1040,745) |
| Yantian | (570,100) | (1050,600) | (1050,650) |
| Singapore | (0,0) | (1060,550) | (1050,550) |

| Return voyage | Singapore | Xiamen | Shanghai |
|---|---|---|---|
| Rotterdam | (570,550) | (640,900) | (640,900) |
| Gdansk | (570,550) | (650,900) | (650,900) |
| Wilhelmshaven | (380,550) | (640,900) | (640,900) |
| Felixstowe | (570,550) | (650,900) | (650,900) |
| Singapore | (0,0) | (660,150) | (650,150) |

Data source: China International Shipping Network.

*5.2. Analysis of Results*

From the optimization results (Table 3), low-speed sailing is the optimal choice under both policies controlling carbon emissions, with the optimal choice of vessel type being a 13,000 TEU ship and an input of seven vessels. An analysis of policies controlling carbon emissions reveals that the carbon tax policy has a large impact on the profit of shipping

companies, while carbon cap-and-trade has a smaller impact on their profit. The carbon tax policy results in a large emission reduction, while that of the carbon cap-and-trade model is small, mainly because the price in the carbon cap-and-trade model is too low, with relatively weak market regulation.

**Table 3.** Analysis of results.

|  | Carbon Cap-and-Trade | Carbon Tax |
| --- | --- | --- |
| daily profit of the fleet | 884,841.6 | 809,853.48 |
| carbon emission | 1851.27 | 1840.09 |
| Carbon emission cost | 5977.27 | 92,004.56 |
| Main engine fuel cost | 168,868.10 | 165,568.10 |
| Fuel cost of auxiliaries | 104,541.10 | 104,541.10 |
| Daily operating cost | 360,766.0 | 360,766.0 |
| port dues | 15,785.71 | 15,785.71 |
| speed | 12.31 | 12.26 |
| speed | 13 | 13 |
| ship type | 7 | 7 |

*5.3. Sensitivity Analysis*

In order to study the influence of carbon tax and carbon trading price changes on enterprises' choice of emission reduction strategies, whether high or low fuel prices affect enterprise emission reduction decisions, and whether ship size is conducive to shipping emission reduction. In this section, the influences of carbon tax, carbon trading price, fuel price, and ship size on carbon emission adjustment policies and fleet profitability are studied through sensitivity analysis.

5.3.1. Sensitivity Analysis of Carbon Tax and Carbon Price

Carbon taxes and carbon cap-and-trade have been adopted as the main policy tools for carbon reduction in most countries emphasizing carbon reduction. With all other parameters unchanged, the model was operated by changing the carbon tax and the carbon price, with the results presented in Figure 5.

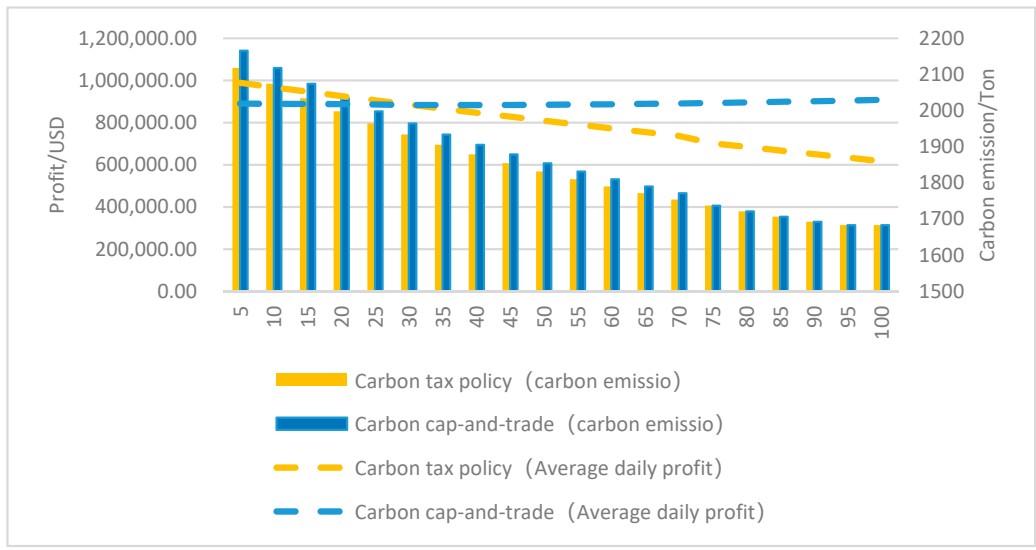

**Figure 5.** Sensitivity analysis of carbon tax and carbon price.

The results suggest that carbon emissions continued to decrease in both the carbon cap-and-trade and carbon tax models with the increase in carbon price and carbon tax. When the carbon price and carbon tax reached $75 USD/ton, the carbon emissions of both models tended to become consistent. From the perspective of company profit, company

profit first declines under the carbon cap-and-trade model. When the carbon price exceeds $50 USD/ton, the company profit starts to rise (as shown in Figure 6). Conversely, under the carbon tax model, company profit decreases, and hence, the comparative analysis of the two models suggests that shipping companies like cap-and-trade, which also allows them to cut emissions.

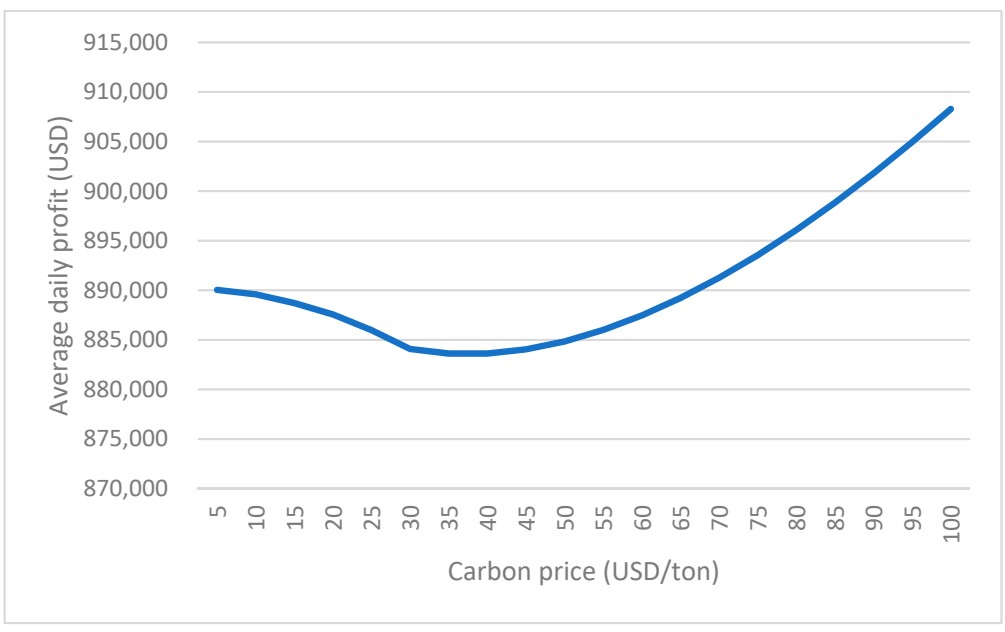

**Figure 6.** Average daily profit curve of the fleet under carbon cap-and-trade policy.

As the carbon market continues to mature and the transaction price continues to increase, results indicate that company profits will increase rather than decrease when companies strive to reduce emissions. For companies, this will provide a greater incentive to optimize transportation to actively reduce emissions. In contrast, the incentive to reduce emissions from a carbon tax is weaker because the tax is compulsory, and the emission reduction behavior driven by a carbon tax is passive.

5.3.2. Changes in Fuel Prices

Currently, most shipping companies use speed changes to control costs. Figure 7 depicts that, in the short term, low oil prices can cut costs for shipping companies. Choosing a high speed when the oil price is low can reduce ship investment or increase the voyage number, while choosing a low speed when the oil price is high can reduce cost. The rising oil price leads to a decrease in ship speed and an increase in ship investment cost. Additionally, Figure 7 indicates that when the oil price drops, the carbon emission of ships is high, so the low oil price will have a negative impact on the carbon emission reduction of ships. For example, the COSCO Shipping Group has invested a substantial amount of money in management software and fuel additives for emission reduction. If low oil prices persist, these investments are largely meaningless from a corporate earnings perspective.

5.3.3. Changes in Vessel Size and Carbon Emissions

Table 4 indicates that larger ships enjoy the dual advantage of lower carbon emissions and lower fuel costs. Adopting larger vessels improves their shipping capacity and significantly reduces the energy consumption of shipping and carbon emissions. Driven by the economy and environmental protection, container ships are transitioning toward larger sizes.

**Table 4.** Optimization results of different vessel types.

| | Carbon Tax | | Carbon Cap-and-Trade | |
|---|---|---|---|---|
| m | 9 | 10 | 9 | 10 |
| Speed | 11.36 | 12.31 | 11.36 | 12.31 |
| Carbon Emissions | 2640.28 | 2162.93 | 2640.28 | 2162.93 |
| Average daily profit of the fleet | 453,292.70 | 559,063.81 | 559,325.02 | 653,162.53 |

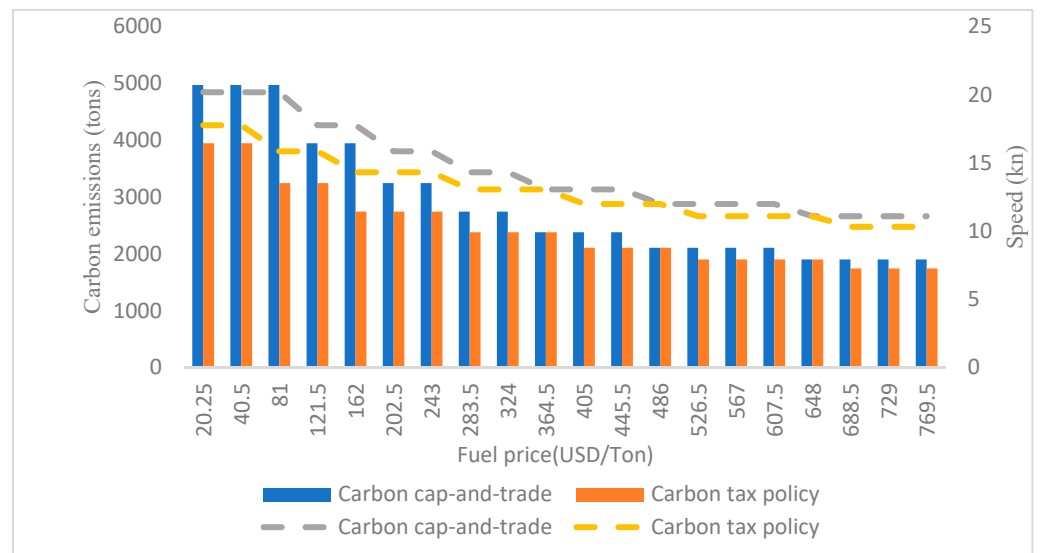

**Figure 7.** The curve of optimization results under different fuel prices.

## 6. Marginal Abatement Cost Analysis

The marginal abatement cost is the cost of each additional unit of $CO_2$ abatement. The classical marginal abatement cost function was proposed by Nordhaus, who provided the relationship between marginal abatement cost and the corresponding emission reduction rate. Combined with the evaluation index of abatement cost per unit $CO_2$ gas proposed by S. E. Magnus, the marginal abatement cost *MAC(ship)* of a ship is calculated with the following equation [37]:

$$MAC(ship) = \Delta C(A)/\Delta CO_2(A) \tag{16}$$

$\Delta C(A)$ is the increase in cost resulting from the adoption of Measure A;
$\Delta CO_2(A)$ is the reduction of $CO_2$ emissions after the adoption of Measure A.

1. *Marginal Abatement Costs of the Carbon Cap-and-Trade Model*

$$\Delta C(A) = C_{Costs\ of\ implementing\ a\ carbon\ Cap-and-Trade\ policy} - C_{No\ carbon\ emission\ limit\ costs}$$
$$\Delta C(A) = \frac{F_0 \cdot D \cdot (P_{HFO} + \lambda_H \cdot p_e)}{168(v^o)^3} v^2_{Carbon\ Cap-and-Trade} - \frac{F_0 \cdot D \cdot P_{HFO}}{168(v^o)^3} v^2_{NO} + \frac{D \cdot (A \cdot (P_{MFO} + \lambda_M \cdot p_e) + C)}{168} \cdot v^{-1}_{Carbon\ Cap-and-Trade}$$
$$- \frac{D \cdot (A \cdot P_{MFO} + C)}{168} \cdot v^{-1}_{NO} - \frac{\lambda_M \cdot p_e \cdot A}{168} \cdot \left( \sum_{i=1}^{N} \frac{Q_i}{l} + Nt_{pil} \right) - \theta \cdot p_e \tag{17}$$

Simplify to obtain

$$\Delta C(A) = \frac{F_0 \cdot D \cdot \lambda_H \cdot p_e}{168(v^o)^3} v^2_{Carbon\ Cap-and-Trade} + \frac{F_0 \cdot D \cdot P_{HFO}}{168(v^o)^3} \left( v^2_{Carbon\ Cap-and-Trade} - v^2_{NO} \right) + \frac{D \cdot A \cdot \lambda_M p_e}{168} \cdot v^{-1}_{Carbon\ Cap-and-Trade}$$
$$+ \frac{D \cdot (A \cdot P_{MFO} + C)}{168} \cdot \left( v^{-1}_{Carbon\ Cap-and-Trade} - v^{-1}_{NO} \right) + \frac{A \cdot \lambda_M \cdot p_e}{168} \cdot \left( \sum_{i=1}^{N} \frac{Q_i}{l} + Nt_{pil} \right) - \theta \cdot p_e \tag{18}$$

$$\Delta CO_2(A) = \frac{\lambda_H \cdot F_0 \cdot D}{168(v^o)^3} \cdot \left(v_{NO}^2 - v_{Carbon\ Cap-and-Trade}^2\right) + \frac{\lambda_M \cdot A \cdot D}{168} \cdot \left(v_{NO}^{-1} - v_{Carbon\ Cap-and-Trade}^{-1}\right) \quad (19)$$

Considering the optimal speed determined without a policy controlling carbon emissions, that is, $v_{null}$ is a known quantity, the model curve was plotted by MATLAB as follows Figure 8:

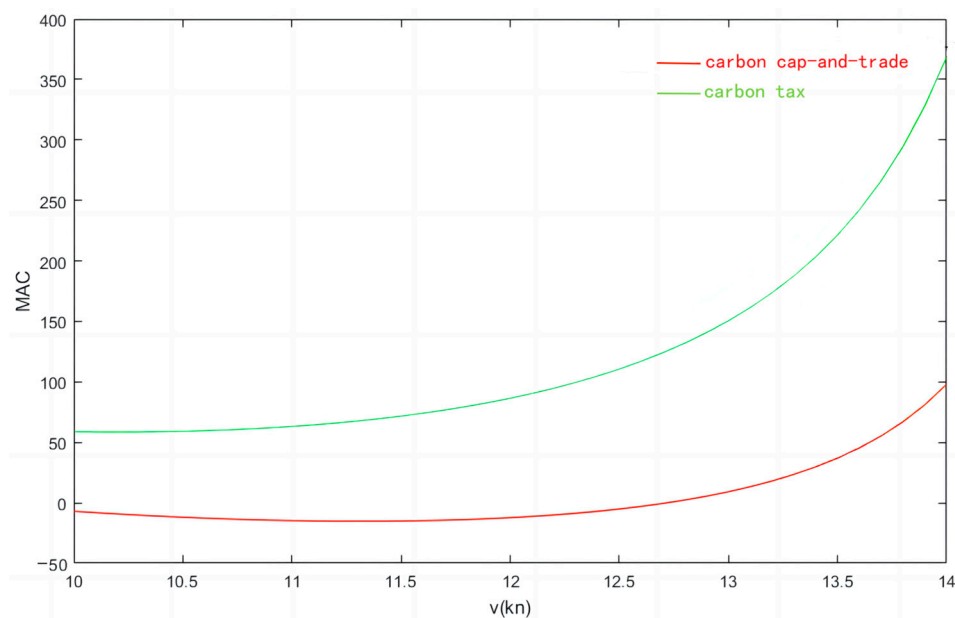

**Figure 8.** Graph of MAC trend.

The carbon cap-and-trade model creates a market for emission rights based on emission control targets, and through carbon trading, an equilibrium market price can be formed such that the efficiency of resource allocation in the shipping market reaches Pareto optimality. Carbon credits are a resource that can bring economic benefits to all shipping companies, with the marginal value of carbon credits varying among them. Shipping companies with higher marginal abatement costs (MAC) can reduce their abatement cost by purchasing carbon credits, while shipping companies with relatively low MAC can earn income by selling carbon credits. From the perspective of MAC, the MAC of shipping companies under the carbon cap-and-trade is smaller than those under the carbon tax policy. In other words, the carbon cap-and-trade makes the emission reduction of ships more profitable and can further encourage shipping companies to reduce emissions.

As shown in the Figures 9–11, the MAC of a ship under the carbon cap-and-trade policy is closely and complexly related to its vessel speed, fuel price, vessel size, and carbon allowance. MAC decreases as the fuel price increases, and the larger the ship type, the larger the MAC.

2. *Marginal Abatement Cost of the Carbon Tax Model*

$$\begin{aligned}
\Delta C(A) &= C_{Cost\ of\ implementing\ carbon\ tax\ policy} - C_{No\ carbon\ emission\ limit\ costs} \\
\Delta C(A) &= \frac{F_0 \cdot D \cdot (\lambda_H \delta + P_{HFO})}{168(v^o)^3} v_{Carbon\ tax}^2 - \frac{F_0 \cdot D \cdot P_{HFO}}{168(v^o)^3} v_{NO}^2 + \frac{D \cdot (A \cdot (P_{MFO} + \lambda_M \delta) + C)}{168} \cdot v_{Carbon\ tax}^{-1} \\
&\quad - \frac{D \cdot (A \cdot P_{MFO} + C)}{168} \cdot v_{NO}^{-1} + \frac{A \cdot \lambda_M \delta}{168} \cdot \left(\sum_{i=1}^{N} \frac{Q_i}{l} + N t_{pil}\right)
\end{aligned} \quad (20)$$

$$\begin{aligned}
\Delta C(A) &= \frac{F_0 \cdot D \cdot \lambda_H \delta}{168(v^o)^3} v_{Carbon\ tax}^2 + \frac{F_0 \cdot D \cdot P_{HFO}}{168(v^o)^3} \left(v_{Carbon\ tax}^2 - v_{NO}^2\right) + \frac{D \cdot A \cdot \lambda_M \delta}{168} \cdot v_{Carbon\ tax}^{-1} \\
&\quad + \frac{D \cdot (A \cdot P_{MFO} + C)}{168} \cdot \left(v_{Carbon\ tax}^{-1} - v_{NO}^{-1}\right) + \frac{A \cdot \lambda_M \delta}{168} \cdot \left(\sum_{i=1}^{N} \frac{Q_i}{l} + N t_{pil}\right)
\end{aligned} \quad (21)$$

$$\Delta CO_2(A) = \frac{\lambda_H \cdot F_0 \cdot D}{168(v^o)^3} \cdot \left(v_{Carbon\ tax}^2 - v_{NO}^2\right) + \frac{\lambda_M \cdot A \cdot D}{168} \cdot \left(v_{Carbon\ tax}^{-1} - v_{NO}^{-1}\right) \tag{22}$$

Considering the optimal speed determined without a policy controlling carbon emissions, that is, $v_{null}$ is a known quantity, the model curve was plotted by MATLAB as follows, Figures 12 and 13:

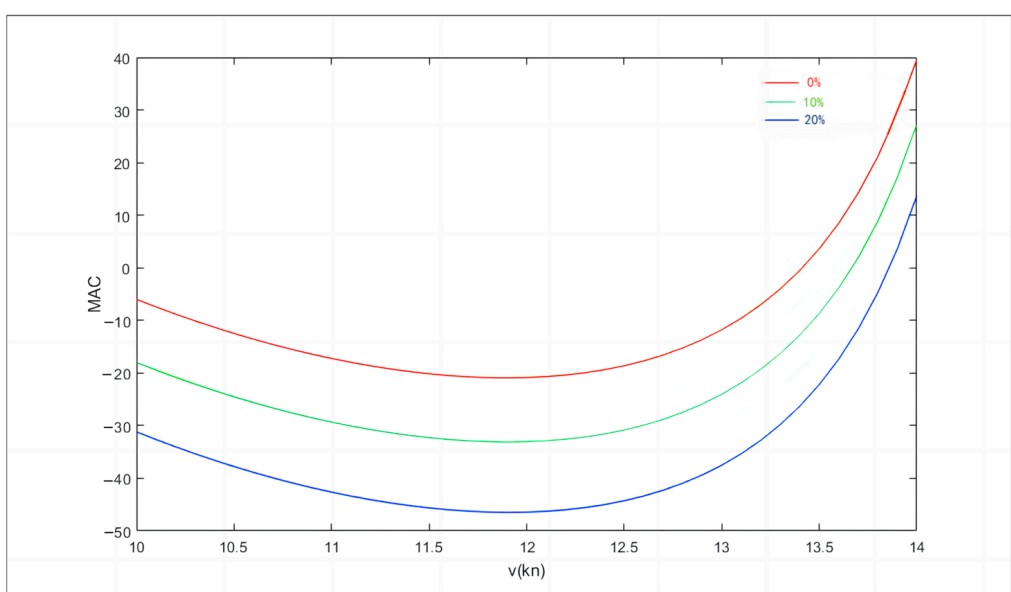

**Figure 9.** Graph of MAC against different fuel prices under the carbon cap-and-trade.

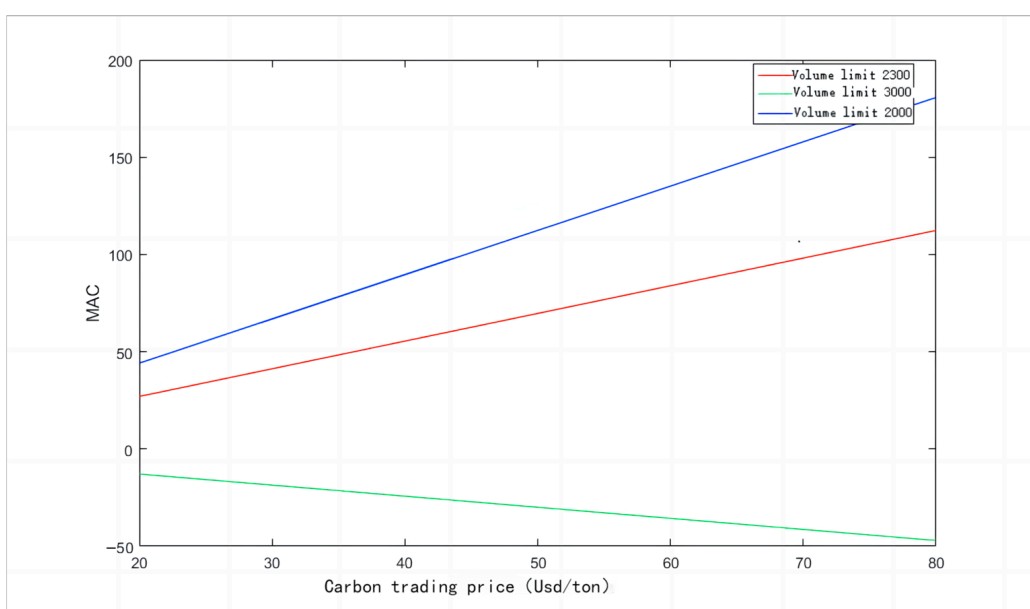

**Figure 10.** Graph of MAC against different carbon allowances.

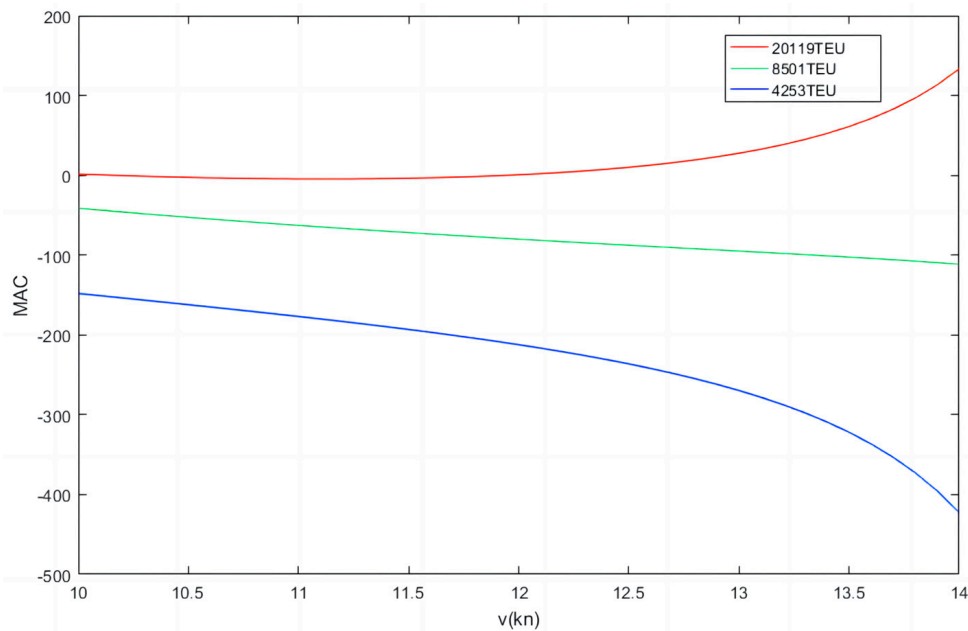

**Figure 11.** Graph of MAC against different vessel types under the cap-and-trade policy.

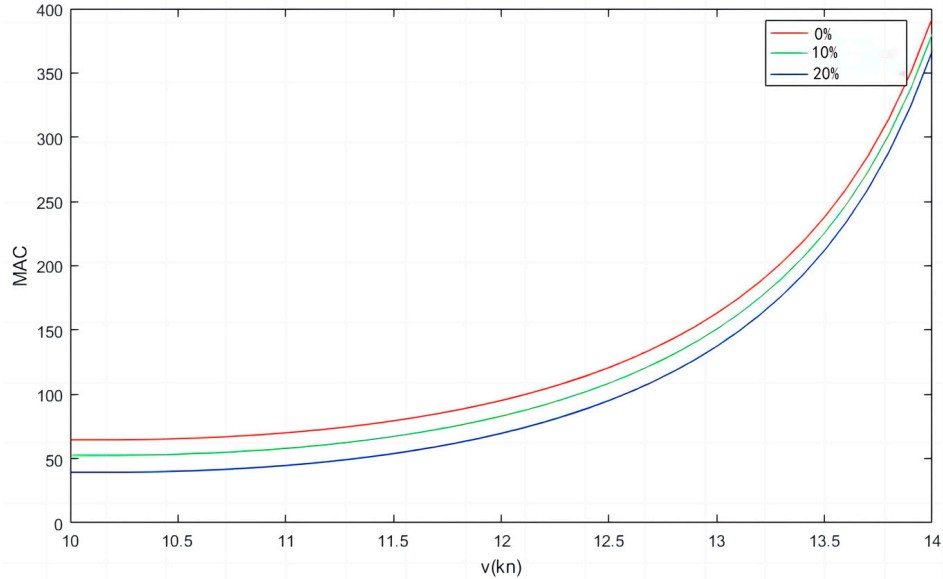

**Figure 12.** Graph of MAC against different fuel prices under the carbon tax policy.

Environmental economic policies mainly follow the Pigovian and Coase approaches. As the carbon tax is essentially an environmental tax, the theoretical basis of the carbon tax policy is grounded on the Pigovian tax theory, which operates by adjusting relative prices and aims to correct environmental externalities by eliminating the difference between private and social prices caused by pollutant emissions. If a Pigovian tax is used to tax carbon emissions from ships, then shipping companies will evaluate their options for action under the effect of economic incentives. When the MAC of the shipping company is lower than the unit carbon tax rate $\delta$, abatement is profitable, and the shipping company can reduce tax payment by abatement. When the MAC of the shipping company is equal to the emission tax rate, the emission tax rate $\delta$ is optimal. When the MAC of the shipping company is greater than the emission tax rate, the abatement policy is less effective or ineffective.

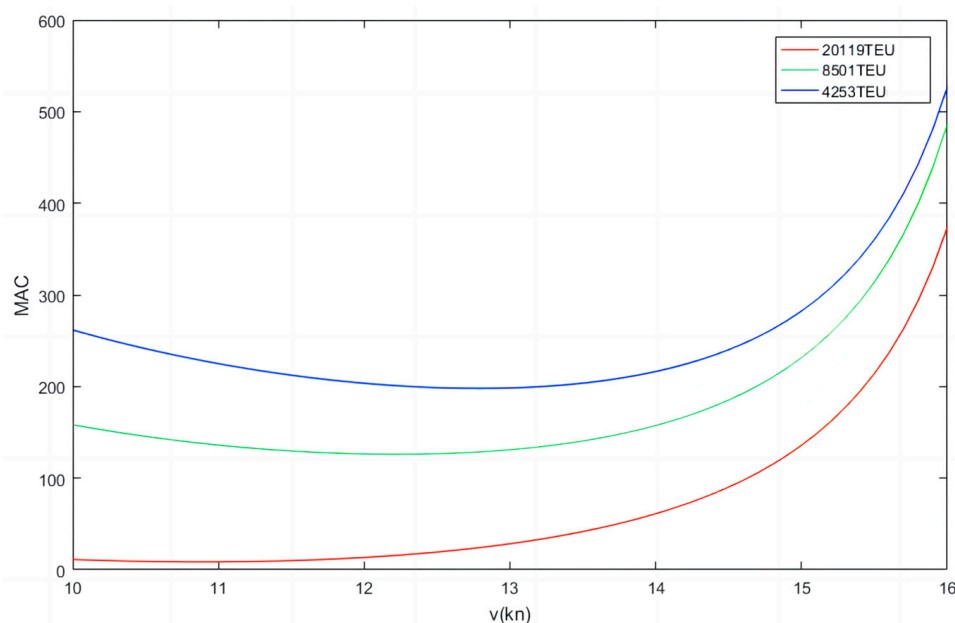

**Figure 13.** Graph of MAC against vessel type under the carbon tax policy.

As presented in the figure, the MAC of a vessel is closely related to its speed and size and the price of fuel. MAC increases with an increase in vessel speed and decreases with an increase in fuel price and vessel size. The lower the fuel price, the higher the marginal cost of abatement, and hence, lower fuel prices negatively impact energy conservation and emission reduction.

*Analysis of the Case Calculation Results*

With the other parameters held constant, the carbon tax (carbon price) was increased for research.

Figure 14 indicates that, under the carbon tax policy, the MAC of carbon emissions with optimized speed was greater than 0 and less than the carbon tax. Optimizing the vessel speed to reduce emissions is, therefore, profitable, and shipping companies are willing to adopt this method to reduce emissions and taxes. Under the carbon cap-and-trade policy, when the trading price is less than USD$50, the marginal emission reduction cost is greater than zero, indicating that the company needs to pay some economic cost to reduce emissions by optimizing the shipping speed. Therefore, the company will not consider reducing emissions by optimizing the shipping speed. Furthermore, the smaller the carbon price, the larger the MAC, and the shipping company will reduce the abatement cost by purchasing carbon credits to maximize profits. When the carbon price is less than USD$50, the marginal emission reduction cost is greater than zero. The shipping company does not need to pay economic costs to optimize the shipping speed and can obtain certain benefits. Therefore, the shipping company is willing to reduce emissions by optimizing the shipping speed. In the case of low carbon trading prices, the carbon cap-and-trade policy has less incentive for shipping enterprises to reduce emissions than the carbon tax policy. With the continuous maturation of the carbon trading market and the continuous increase in carbon trading price, the carbon cap-and-trade policy will be welcomed by shipping enterprises that actively reduce emissions, and the enthusiasm of shipping companies to reduce emissions will be greatly enhanced. From the perspective of the long-term development of the green shipping market, a carbon cap-and-trade policy is better than a carbon tax policy and more in line with the sustainable development of shipping.

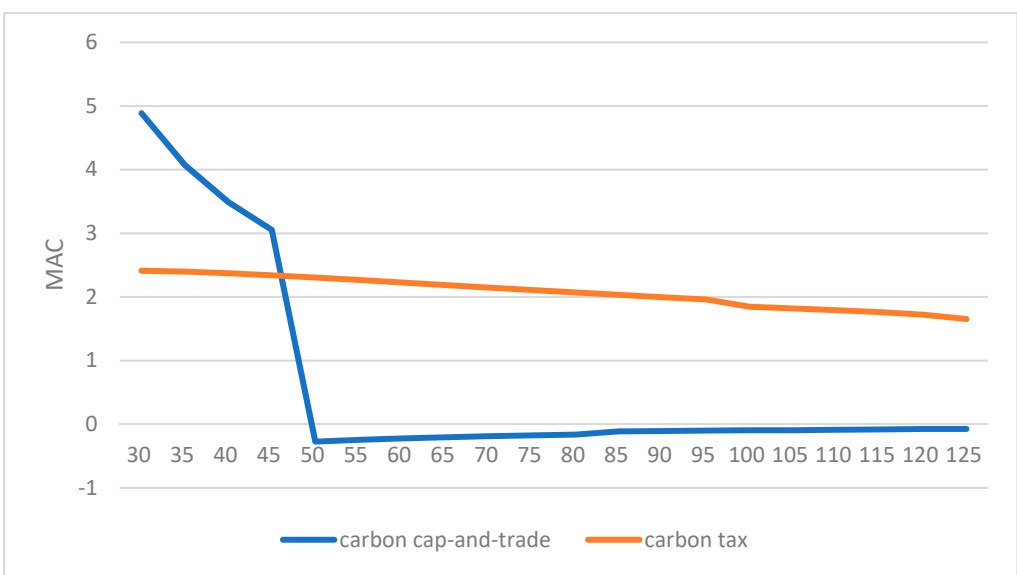

**Figure 14.** Graph of calculated MAC results under the carbon tax and carbon cap-and-trade policies.

## 7. Conclusions

This study considers different carbon emission policies to construct an optimization model for container liner shipping, design an improved Whale Swarm Algorithm to solve related issues, and use the marginal carbon abatement cost method to analyze the deep-seated reasons for the optimization of liner shipping according to different carbon emission policies, thereby revealing the underlying reasons of emission-reduction decisions. The following conclusions were derived:

- The nonlinear optimization model constructed in this paper and the improved Whale Swarm Algorithm designed in this paper can well solve the container transportation optimization problem under different carbon emission policies and can provide enterprises with better ship speed, ship investment, and ship type selection reference.
- Both kinds of carbon emission regulation policies will lead to a reduction in the profits of liner transport companies, a reduction in average speed, and a reduction in carbon emissions. The carbon tax model has the greatest impact on the profits of shipping companies, and carbon cap-and-trade is easier to obtain support from enterprises. However, the current carbon trading price is too low, and its market regulation effect is weak.
- Under the carbon tax policy, the marginal cost of reducing carbon emissions by means of optimizing shipping speed is less than that of a carbon tax, and the emission reduction method of optimizing shipping speed is profitable. The shipping company is willing to reduce emissions by this method to reduce tax. Under the condition that the current carbon trading market is not mature enough and the trading price is low, compared with the carbon tax model, the carbon cap-and-trade model has less incentive for shipping enterprises to reduce emissions. As the carbon trading market continues to mature and the carbon trading price continues to rise, shipping companies will greatly improve their enthusiasm for reducing emissions. From the perspective of the emission reduction effect and enterprise enthusiasm, the carbon cap-and-trade model is better than the carbon tax model, which is more in line with the sustainable development of shipping.
- The implementation of a carbon cap-and-trade policy or carbon tax policy is closely and complexly related to carbon trading price, carbon tax rate, fuel price, and ship size, and there are uncertainties. The uncertainty caused by these factors makes the uncertainty of the implementation effect of each policy should arouse the attention of relevant decision-makers. The optimal measures should be formulated under the premise of fully understanding the uncertainty factors.

- Policy suggestions: (1) It is the general trend to formulate carbon emission policies reasonably and achieve carbon emission reduction through dual means of regulation and market; (2) In the case of immature carbon trading markets, carbon taxes are more flexible, allowing enterprises to respond to market signals in the most economical way, choose between paying taxes and reducing carbon emissions, and significantly eliminate the environmental externalities of carbon emissions. However, the imposition of a carbon tax will increase the financial burden of shipping companies and inhibit the development of the shipping market to some extent. Therefore, the setting of a carbon emission tax rate is particularly important. We can refer to the current marginal emission reduction cost MAC of the shipping industry to set the tax rate in line with the actual development. (3) The emission reduction effect of carbon cap-and-trade in total cap control is clear. No matter what behavior choice shipping companies adopt, the total emission reduction control target in the equilibrium shipping market is constant and can promote the development of low-carbon production technology. Therefore, the establishment of a carbon emission trading market in the shipping market should be accelerated.

This study does not consider the weather factor, the weather has a certain impact on ship navigation, and the impact has significant randomness, real-time and dynamic. In the future, environmental factors can be taken into account to establish an optimization model of variable speed transportation affected by random factors.

**Author Contributions:** Conceptualization and methodology, X.L., X.Z. and Q.T.; writing—original draft preparation, X.L. and Q.T.; writing—review and editing, X.L.; software, X.L. and X.Z.; supervision, X.Z. All authors have read and agreed to the published version of the manuscript.

**Funding:** This research was funded by the Marine Economy Development Special Fund Project of Guangdong Province—Guangdong Natural Resources Cooperation [2020], grant number 071.

**Institutional Review Board Statement:** Not applicable.

**Informed Consent Statement:** Not applicable.

**Data Availability Statement:** Not applicable.

**Conflicts of Interest:** The authors declare no conflict of interest.

**Description of the Variables:**

Decision variables

| | |
|---|---|
| $n_m$ | is the number of $m$ vessels assigned; |
| $x_m$ | is variable 0, 1, and its value is 1 when $m$ vessel is configured on the route and 0 otherwise; |
| $V_m$ | is ship speed (kn); |

Port-related parameters

| | |
|---|---|
| $D$ | Total route distance (kn); |
| $q_{ij}$ | is the weekly freight volume between ports $j$ and $i$ (*TEU*); |
| $N$ | Number of ports on the route; |
| $r_{ij}$ | is the freight rate between ports $j$ and $i$ (USD\$/TEU). |
| $t_{pil}$ | is the sum of berthing and unberthing time of the ship in and out of the port (hour); |
| $l$ | The overall average handling efficiency of the port (TEU/hour); |
| $C_m$ | Fixed daily rate of m-vessel (USD\$/day); |
| $C^l$ | Port charges for loading containerized cargo (USD\$/TEU); |
| $C^u$ | Port charges for unloading containerized cargo (USD\$/TEU); |
| $G_m^0$ | is the entry fee of the $m$ vessel (USD/time), including berthing fee, mooring and unmooring fee, port clearance fee and other charges incurred in port; |

Ship-related parameters

| | |
|---|---|
| M | is the set of ship types. |
| $Q_{CO_2}$ | Weekly average $CO_2$ emissions of the fleet (tons). |
| $C_{CO_2}$ | Costs of carbon emissions. |
| $f^{day}$(tp) | is the average daily profit of the fleet (USD\$). |
| $F_m$ | Main engine daily fuel consumption of m-vessel at designed speed (t/day). |
| $A_m$ | Auxiliaries daily fuel consumption of m-vessel at designed speed (t/day). |
| $P_{IFO}$ | Heavy crude oil prices (USD\$/ton). |
| $P_{VLSFO}$ | Very low sulfur fuel oil prices (USD\$/ton). |
| $\lambda$ | Carbon conversion factor of ships. |
| $T$ | The expression of the total voyage time (day). |
| $Y_k$ | The total amount of cargo (tons) on segment $k$. |
| $Q_i$ | is the loading and unloading of port $i$. |
| $C^{op}$ | The weekly operating costs of the fleet (USD\$). |
| $C^P$ | The average port charges per week (USD\$). |

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
