# Peer review of "Container Shipping Optimization under Different Carbon Emission Policies: A Case Study"

_sustainability, doi:10.3390/su15108388_

Round 1

Reviewer 1 Report

comments in the attached file 

comments in the attached file

Author Response

All comments have been accepted and changes have been made. See the revised draft for details

Reviewer 2 Report

Dear Editor of Sustainability:

The manuscript by Xiangang Lan et al., Liner Transportation Optimization Under Different Carbon Emission Policies: A Real Case Study. The contents of this manuscript have some good science and concepts, but the paper needs improvement before it is accepted for publication. I recommend that this manuscript be published in "Sustainability" after revision. My detailed comments are as follows:

Reviewer comments:

1.    There are lots of English grammar errors that should be corrected in the manuscript.

2.    The author should improve the quality of figures such as Figures 5, 6, 8, 9, 10, 11, and 14.

3.    The authors should provide pictures about CO2 emissions in different ways in the introduction section in order to be better seen and understandable for readers.

4. In order to increase the quality of the paper, the authors should write sentences about “the role of chemicals, fabricated materials, and chemical industry sectors in the global CO2 emissions” in the introduction section and cite some papers which Include:

https://doi.org/10.15157/EIL.2023.1.1.28-45,

https://doi.org/10.15157/EIL.2023.1.1.10-27

https://doi.org/10.1016/j.foodchem.2021.129042

 https://doi.org/10.1002/aoc.5989

Best Regards

There are lots of English grammar errors that should be corrected in the manuscript.

Author Response

  1. Revised
  2. Revised
  3. This paper studies the policy of carbon emission regulation

  4. This paper discusses the problem of carbon emission from ships

Reviewer 3 Report

1. This paper considers the policy of carbon emission quota and tax, takes ship type, quantity and ship speed as input, and uses Whale Swam Algorithm to research Liner Transportation Optimization problem. The research method proposed by the author is innovative, and the research results have certain reference value for practical application.

2. There are many factors affecting Liner Transportation Optimization, and there are also great uncertainties. In this paper, the author sets six hypothetical conditions, which are actually difficult to meet at the same time, which affects the research value of this paper. It is hoped that the author can further consider these hypotheses and put forward some hypotheses close to the actual situation to make the research results more practical. For example, some meteorological conditions in ship navigation will seriously affect ship speed; In some special routes, such as canals and sea areas near harbors, the navigation of ships is greatly affected.

3. For objective function optimization, a variety of optimization algorithms can be realized. Whale Swam Algorithm is a kind of optimization Algorithm proposed in recent years. Based on this algorithm, some scholars put forward a variety of improved Whale Swam algorithms. In this study, the author also proposed an improved Whale Swam Algorithm, "especially considering that existing algorithms have proven to be less effective in solving the model." Please explain the necessity of using Whale Swam Algorithm for this research. Problems with other algorithms; improved Whale Swam Algorithm in this article What improvements have been made to the Whale Swam Algorithm? If there is no substantial innovation in the improved Whale Swam Algorithm, it is suggested to simplify Section 5 of the article.

4. In the conclusion of the article, the authors argue that "This study examined the optimization of liner shipping with the carbon cap-and-trade policy, designed an improved Whale Swarm Algorithm (WSA) combined with empirical analysis to verify the effectiveness of the model and algorithm, and conducted a comparative study using the marginal carbon abatement cost method ". However, from the research of this article, the results of "effectiveness" are valid only under some assumptions that are quite different from the actual situation.

5. the first part of the conclusion is "Against the background of policies controlling carbon emissions, low-speed sailing is the optimal choice for liner shipping "algorithm reasonable? Please reconsider.

Author Response

1.Considering the weather factor will make the model very complicated. Many scholars have conducted researches on the optimization of ship speed considering the weather factor. However, this paper focuses on the research on the regulation policy of carbon emission, which can ignore the weather factor;

2.The algorithm design part has been simplified;

3.The conclusion has been revised

Reviewer 4 Report

This study investigates the optimization model of liner shipping under different carbon emission policies and evaluates the impact of carbon tax policy and carbon cap-and-trade policy on the optimization results. This study addresses the current hot issues of container shipping research and provides some reference for carbon emission reduction.

General Remarks

The name of the variable indicated by the horizontal axis in Figure 10 uses an inappropriate language type, and the legends in Figure 12 and Figure 13 have the same problem. Please check all the figures and make corrections.

All variables in the equation should be explained, please check all equations and make corrections.

Section remarks

Section 1 - Introduction

I liked how you built importance and point out the focus of your study. However, I'd like to see how you express the research questions of your study. To be more specific in your research study aims, trying to answer some specific questions to help the reader understand your study.

Section 2 - Literature Review

Literature review is not a simple listing of the literature; it needs to synthesize the lineage of the development of a particular study, such as the proposal, improvement, and development of a particular method. The literature review section must be re-summarized and condensed.

Section 6 – Case study

Can you provide some optimization suggestions based on your research results? You have analyzed the deep-seated reasons for optimizing liner shipping under two carbon reduction policies, however, it would complete your study if you propose optimization solutions for different carbon reduction policies tailored to the liner shipping situation.

Section 7 – Conclusions

Acknowledge the limitations of your data and analysis, and provide suggestions for further research.

Author Response

1.General Remarks:Revised

2.All variables in the equation should be explained, please check all equations and make corrections:Revised

3.Section 1 - Introduction:Revised

4.Section 2 - Literature Review:Revised

5.Section 6 – Case study:Revised(see line 630-646)

6.Section 7 – Conclusions:Revised(see line 647-650)

Round 2

Reviewer 1 Report

All raised concerns have been addressed satisfactory. Congrats and the manuscript is recommended for publishing.

Still some minor revisions required. 

Reviewer 4 Report

The article is much improved in this revised version.

The authors correctly addressed the revision.